# Solving Hard Analogy Questions with Relation Embedding Chains

**Nitesh Kumar** and **Steven Schockaert**
Cardiff NLP, School of Computer Science and Informatics
Cardiff University, United Kingdom
{kumarn8,schockaerts1}@cardiff.ac.uk

## Abstract

Modelling how concepts are related is a central topic in Lexical Semantics. A common strategy is to rely on knowledge graphs (KGs) such as ConceptNet, and to model the relation between two concepts as a set of paths. However, KGs are limited to a fixed set of relation types, and they are incomplete and often noisy. Another strategy is to distill relation embeddings from a fine-tuned language model. However, this is less suitable for words that are only indirectly related and it does not readily allow us to incorporate structured domain knowledge. In this paper, we aim to combine the best of both worlds. We model relations as paths but associate their edges with relation embeddings. The paths are obtained by first identifying suitable intermediate words and then selecting those words for which informative relation embeddings can be obtained. We empirically show that our proposed representations are useful for solving hard analogy questions.[1]

## 1 Introduction

Many applications rely on Knowledge graphs (KGs) to model the relationship between concepts. For instance, KGs have been used to characterise how an answer candidate is related to the concepts that appear in a question (Yasunaga et al., 2021; Zhang et al., 2022b) and to help interpret visual scenes (Gu et al., 2019). In such applications, relations are modelled as KG paths, which has two key advantages: (i) we can easily inject domain-specific knowledge and (ii) the interpretable nature of KG paths offers a degree of transparency. But KGs also have important drawbacks. They use a fixed set of relation types, which may not be fine-grained enough. KGs are also inevitably incomplete and, especially in the case of commonsense KGs such as ConceptNet (Speer et al., 2017), noisy.

There is also a tradition in Natural Language Processing (NLP) to model relations as embeddings. For instance, DIRT (Lin and Pantel, 2001) and LRA (Turney, 2005) are early examples of methods which used vectors to represent the relation between two words. More recently, relation embeddings have been obtained using fine-tuned language models. For instance, RelBERT (Ushio et al., 2021a) has achieved state-of-the-art results on several analogy datasets using a fine-tuned RoBERTa-large model (Liu et al., 2019). The strengths and weaknesses of relation embeddings are complementary to those of KGs: relation embeddings lack transparency and cannot easily incorporate external knowledge, but they are capable of capturing subtle differences, making it possible to encode relational knowledge in a way that is considerably richer than what can be achieved with KGs. A final drawback of relation embeddings is that they are best suited for concept pairs that have a clear and direct relationship. For instance, they can encode the relation between *umbrella* and *rain*, but are less suitable for encoding the relation between *umbrella* and *cloudy* (e.g. if it is cloudy, there is a chance of rain, which means that we might take an umbrella).

In this paper, we propose a hybrid approach which combines the advantages of KGs with those of relation embeddings. The idea is conceptually straightforward. We represent relations as paths, where nodes correspond to concepts, but rather than associating edges with discrete types, we label them with relation embeddings. We will refer to such paths as *relation embedding chains*. Clearly, this approach allows us to model indirect relationships, while keeping the flexibility of embeddings, as well as some of the interpretability of KG paths.

We are still faced with the challenge of selecting suitable paths. KGs are insufficient for this purpose, given their noisy and incomplete nature. Our solution relies on the following idea: to decide whether $a \rightarrow x \rightarrow b$ is a suitable path for

---

[1]Source code to reproduce our experimental results and the model checkpoints are available in the following repository: https://github.com/niteshroyal/SolvingHardAnalogyQuestions.

modelling the relationship between $a$ and $b$, what matters is whether we have access to an informative relation embedding for the word pairs $(a, x)$ and $(x, b)$. Motivated by this, we first develop a classifier to predict whether a given RelBERT embedding is informative or not. We then generate possible relational paths, using ConceptNet as well as standard word embeddings, and filter these paths based on the informativeness classifier.

While relation embedding chains are expressive, we sometimes need a simpler representation, especially in unsupervised settings. We therefore also study how the information captured by relation embedding chains can be summarised as a single vector, without relying on task-specific supervision.

To evaluate the usefulness of relation embedding chains, we focus on word analogy questions. Given a query word pair (e.g. *word:language*) and a set of candidate word pairs, the task is to select the most analogous candidate (e.g. *note,music*). We show that relation embedding chains are well-suited for answering hard analogy questions.

## 2 Related Work

Modelling analogies has been a long-standing topic of interest in NLP (Turney, 2005, 2012). Recent years have seen a significant increase in interest in this topic, fuelled by the success of large language models (LLMs). For instance, Bhavya et al. (2022) used LLMs for generating explanations of scientific concepts involving analogies, while Sultan and Shahaf (2022) used LLMs for identifying analogies between procedural texts. However, most work has focused on modelling analogies between word pairs (Ushio et al., 2021b,a; Chen et al., 2022; Czinczoll et al., 2022a; Li et al., 2023; Yuan et al., 2023), which is also the setting we consider in this paper. We focus in particular on RelBERT (Ushio et al., 2021a), which is the state-of-the-art on several benchmarks. RelBERT is a RoBERTa-large model that was fine-tuned on the relational similarity dataset from SemEval 2012 Task 2 (Jurgens et al., 2012). Given a word pair, RelBERT computes a relation embedding by feeding that word pair as input to the fine-tuned RoBERTa model and averaging the output embeddings.

The use of KG paths for modelling relations between concepts has also been extensively studied. For instance, Boteanu and Chernova (2015) proposed the use of ConceptNet paths for explaining why two word pairs are analogous. Zhou et al.

(2019) highlighted the noisy nature of many ConceptNet paths. To address this issue, they trained a system to predict path quality based on crowd-sourced judgments of naturalness. More recently, ConceptNet paths have been used to provide external knowledge to NLP systems, for instance in question answering systems (Lin et al., 2019; Yasunaga et al., 2021; Zhang et al., 2022b; Jiang et al., 2022; Sun et al., 2022). In such cases, a Graph Neural Network is typically used to process the paths, and the resulting representations are then integrated with those obtained by a language model. An important distinction with our work is that we focus on unsupervised settings. KG paths are especially helpful for question answering over KGs. However, most approaches rely on matching chains of discrete relation types (Das et al., 2022; Zhang et al., 2022a). Chains of relation embeddings, as we study in this paper, make it possible to model finer relationships.

## 3 Scoring RelBERT Embeddings

Given a word pair $(a, b)$, we write $\mathbf{r_{ab}} \in \mathbb{R}^d$ for the corresponding RelBERT embedding. While RelBERT can provide a relation embedding for any pair of words, not all these embeddings are equally informative. In particular, we would like to distinguish embeddings $\mathbf{r_{ab}}$ which encode a specific relationship from embeddings that rather reflect the lack of any (known) relationship between $a$ and $b$. Unfortunately, the relation embeddings of different unrelated word pairs are typically not similar.

To address this, we want to train a classifier to predict whether a given relation embedding $\mathbf{r_{ab}}$ is informative or not. However, this requires access to a set of related word pairs *Pos* and a set of unrelated word pairs *Neg*. The set *Neg* can simply be constructed by choosing random word pairs. While we may occasionally end up with word pairs that are actually related in some way, the probability that this is the case for randomly chosen words is sufficiently low for this not to be problematic. However, constructing the set *Pos* is less straightforward, since we are not looking for words that have a particular relation, but rather for words that have *any* kind of (clear) relationship. If the examples are not sufficiently diverse, then our classifier will simply detect whether $\mathbf{r_{ab}}$ corresponds to one of the relations that were considered during training. One of the few relevant large-scale resources we have at our disposal is ConceptNet. However, during

initial experiments, we quickly found ConceptNet to be too noisy for this purpose. We therefore instead relied on GPT-4[2] to generate a diverse set of around 11,000 positive examples. The prompt we used to obtain a sufficiently diverse set of examples is shown in the Appendix D.

To create a set of negative examples, of the same size, we corrupted the positive examples. We then trained a logistic regression classifier on the resulting training set. Throughout this paper, we will write $inf(\mathbf{r_{ab}})$ for the probability that $\mathbf{r_{ab}}$ expresses an informative relationship, according to this classifier. We will refer to this value as the informativeness of the embedding $\mathbf{r_{ab}}$.

## 4 Connecting Concepts

The relationship between two concepts $a$ and $b$ is sometimes easiest to explain using an intermediate concept $x$. For instance, *umbrella* is related to *cloudy* because (i) an *umbrella* protects against *rain* and (ii) *rain* is only possible if it is *cloudy*. A natural strategy for identifying such intermediate concepts is to find paths of length two connecting the target words $a$ and $b$ in ConceptNet. However, the coverage of ConceptNet is limited, and many relevant intermediate concepts might not be found in this way. One possible solution would be to consider a sequence of intermediate concepts, and model the relation between $a$ and $b$ in terms of a path $a \rightarrow x_1 \rightarrow ... \rightarrow x_n \rightarrow b$. However, longer ConceptNet paths are often too noisy to be useful. Moreover, in practice a single intermediate concept is usually sufficient, so rather than considering longer paths, we propose two strategies to find additional links between concepts based on word embeddings. They are illustrated in Figure 1.

**Missing Link Prediction** We add missing links based on the informativeness classifier from Section 3. Specifically, we first use a standard word embedding to find the top-$k$ most similar words to $a$, with e.g. $k = 500$. For each of these words $y$, we add a link between $a$ and $y$ if $inf(\mathbf{r_{ay}}) > 0.75$.[3] We add missing links from $b$ in the same way. We will refer to this strategy as *missing link prediction*. Note that this strategy not only finds synonyms or morphological variations of words but also often

[2] https://openai.com/research/gpt-4
[3] Since we focused on unsupervised settings in this paper, we cannot tune this value. Compared to a threshold of 0.5, the choice of 0.75 reflects the idea that we want to be selective: if in doubt it seems better not to add the link.

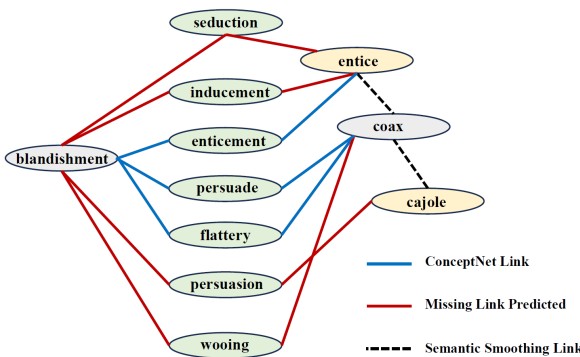

Figure 1: For the concept pair (*blandishment, coax*), the integration of missing link prediction and semantic smoothing strategies identifies *seduction, inducement, enticement, persuade, flattery, persuasion,* and *wooing* as intermediate concepts. The concepts *entice* and *cajole* used for smoothing are not considered intermediate concepts; they merely help in inducing additional intermediate concepts. If only ConceptNet were used to determine intermediate concepts, *flattery* and *persuade* would be the only ones identified.

finds words that are related in different ways. For instance, some links that are not present in ConceptNet and have been added using this approach include: (*dog, owners*), (*cashier, grocery store*), (*helium, noble gases*), (*drug trafficking, illegal*), and (*disinfectant, sterilization*). Such links clearly enrich ConceptNet in a non-trivial way.

**Semantic Smoothing** For the second strategy, we only consider the top-5 neighbours, noting that these are often either synonyms or morphological variations of the same word (e.g. *cloud* and *cloudy*). Specifically, rather than only considering $x$ as an intermediate concept if $x$ is connected to both $a$ and $b$, we now consider $x$ as an intermediate concept as soon as it is connected to one of the 5 nearest neighbours of $a$ (or $a$ itself) and one of the 5 nearest neighbours of $b$ (or $b$ itself). We will refer to this second strategy as *semantic smoothing*.

## 5 Condensing Relation Embedding Chains

The strategy from Section 4 allows us to identify intermediate concepts $x_1, ..., x_n$ that can help to describe the relation between two given words $a$ and $b$. Each intermediate concept $x_i$ corresponds to a chain $\mathbf{r_{ax_i}}; \mathbf{r_{x_ib}}$ of two relation embeddings. These *relation embedding chains* can encode relationships in a fine-grained way, but in practice we often need a more compact representation. We therefore train a simple model to summarise a set of relation em-

bedding chains $\{\mathbf{r_{ax_1}}; \mathbf{r_{x_1b}}, ..., \mathbf{r_{ax_n}}; \mathbf{r_{x_nb}}\}$ for a given concept pair $(a, b)$ as a single vector. Our model has the following form:

$$\mathbf{s_{ab}} = \psi\Big(f\big(\phi(\mathbf{r_{ax_1}}, \mathbf{r_{x_1b}}), ..., \phi(\mathbf{r_{ax_n}}, \mathbf{r_{x_nb}})\big)\Big)$$

The function $\phi : \mathbb{R}^d \times \mathbb{R}^d \to \mathbb{R}^m$ is intuitively doing a kind of relational composition: it aims to predict a relation embedding for the pair $(a, b)$ from the relation embeddings $\mathbf{r_{ax_i}}$ and $\mathbf{r_{x_ib}}$. These predicted embeddings are then combined using a pooling function $f : \mathbb{R}^m \times ... \times \mathbb{R}^m \to \mathbb{R}^m$. Finally, the resulting vector is mapped back to a $d$-dimensional embedding using the decoder $\psi : \mathbb{R}^m \to \mathbb{R}^d$.

For the decoder $\psi$, we simply use a linear layer. For the pooling function $f$, we experimented with sum-pooling and max-pooling. We found the result to be largely insensitive to this choice. Throughout this paper, we will therefore fix $f$ as summation. Finally, the composition function $\phi$ is implemented as follows:

$$\phi(\mathbf{r_{ax}}, \mathbf{r_{xb}}) = \mathsf{GeLU}(\mathbf{A}(\mathbf{r_{ax}} \oplus \mathbf{r_{xb}}) + \mathbf{b}) \quad (1)$$

where we write $\oplus$ for vector concatenation, $\mathbf{A}$ is a matrix, and $\mathbf{b}$ is a bias vector.

**Training** Our focus is on solving analogy questions, which is an unsupervised problem. In the absence of task-specific training data, we train the model to predict the RelBERT embedding $\mathbf{r_{ab}}$, using the following loss:

$$\mathcal{L} = -\sum_{(a,b)} \cos(\mathsf{s_{ab}}, \mathsf{r_{ab}}) \quad (2)$$

where $\mathsf{s_{ab}}$ is the vector predicted from the relation embedding chains, and $\mathsf{r_{ab}}$ is the RelBERT embedding. The sum in (2) ranges over concept pairs $(a, b)$ from ConceptNet for which the informativeness $inf(\mathbf{r_{ab}})$ is sufficiently high. This is important to ensure that the model is not trained on a noisy supervision signal. Specifically, we only considered concept pairs $(a, b)$ for which $inf(\mathbf{r_{ab}}) > 0.75$.

Note that while we train the model on concept pairs $(a, b)$ which have an informative RelBERT embedding, our aim is to use this model for pairs for which this is not the case. We thus rely on the assumption that a composition model which is trained on word pairs with informative RelBERT vectors will generalise in a meaningful way to word pairs for which this is not the case.

## 6 Solving Analogy Questions

Analogy questions involve a query pair $(a, b)$ and a number of candidate answers $(x_1, y_1), ..., (x_k, y_k)$. The aim is to select the candidate $(x_i, y_i)$ whose relationship is most similar to that of the query pair. When using RelBERT, we simply choose the pair $(x_i, y_i)$ for which $\cos(\mathbf{r_{ab}}, \mathbf{r_{x_iy_i}})$ is maximal.

**Identifying Hard Analogy Questions** Our key hypothesis is that the informativeness of the Rel-BERT vectors, as predicted by the classifier from Section 3, can tell us whether using RelBERT vectors is a reliable strategy for a particular analogy question. The lower the informativeness of $\mathbf{r_{ab}}$ or $\mathbf{r_{x_iy_i}}$, with $(a, b)$ the query and $(x_i, y_i)$ the chosen answer candidate, the less confident we can be about the chosen answer. We define our confidence in RelBERT's prediction as follows:

$$conf(a, b, x, y) = min(inf(\mathbf{r_{ab}}), inf(\mathbf{r_{xy}})) \quad (3)$$

with $(a, b)$ the query pair and $(x, y)$ the candidate selected by RelBERT. In cases where the confidence is low, we aim to improve the prediction by taking advantage of relation embedding chains.

**Condensed Relation Chain Comparison** We can straightforwardly solve analogy questions using condensed relation chains. In particular, we use the model from Section 5 to obtain relation embeddings $\mathbf{s_{xy}}$ for the query and the candidate pairs. We then proceed in the same way as with RelBERT, choosing the answer candidate $(x_i, y_i)$ for which $\cos(\mathbf{s_{ab}}, \mathbf{s_{x_iy_i}})$ is maximal.

**Direct Relation Chain Comparison** We can also solve analogy questions by using relation chains directly. Let $c_1, ..., c_u$ be the intermediate concepts for the query pair $(a, b)$ and let $z_1, ..., z_v$ be the intermediate concepts for an answer candidate $(x, y)$. Then we evaluate the compatibility $comp(a, b, x, y)$ of this candidate as follows:

$$\sum_{i=1}^{u} \max_{j=1}^{v} sim(\mathbf{r_{ac_i}}; \mathbf{r_{c_ib}}, \mathbf{r_{xz_j}}; \mathbf{r_{z_jy}})$$

The idea is that when $(a, b)$ and $(x, y)$ are analogous, for most relation chains $\mathbf{r_{ac_i}}; \mathbf{r_{c_ib}}$ connecting $a$ and $b$ there should exist a similar relation chain $\mathbf{r_{xz_j}}; \mathbf{r_{z_jy}}$ connecting $x$ and $y$. The similarity between relation chains can be evaluated as follows:

$$sim_1(\mathbf{r_{ac_i}}; \mathbf{r_{c_ib}}, \mathbf{r_{xz_j}}; \mathbf{r_{z_jy}}) \quad (4)$$
$$= min(\cos(\mathbf{r_{ac_i}}, \mathbf{r_{xz_j}}), \cos(\mathbf{r_{c_ib}}, \mathbf{r_{z_jy}}))$$

In other words, two relation chains are similar if their first component is similar and their second component is also similar. Our default configuration uses this approach. However, this may be too strict. For instance, suppose that $\mathbf{r_{c_i b}}$ and $\mathbf{r_{z_j y}}$ capture the *located-at* relation, while $\mathbf{r_{ac_i}}$ captures *part-of* and $\mathbf{r_{xz_j}}$ captures *is-a*. Then the relation chains $\mathbf{r_{ac_i}}; \mathbf{r_{c_i b}}$ and $\mathbf{r_{xz_j}}; \mathbf{r_{z_j y}}$ both essentially encode that $a$ is located at $c$, but they would not be considered to be similar according to (4). To allow such relation chains to be identified as being similar, we will also consider the following alternative:

$$sim_2(\mathbf{r_{ac_i}}; \mathbf{r_{c_i b}}, \mathbf{r_{xz_j}}; \mathbf{r_{z_j y}}) \qquad (5)$$
$$= \cos(\psi(\phi(\mathbf{r_{ac_i}}; \mathbf{r_{c_i b}})), \psi(\phi(\mathbf{r_{xz_j}}; \mathbf{r_{z_j y}})))$$

with $\phi$ and $\psi$ the composition function and decoder of the model for condensing relation chains. Finally, we will also consider a baseline strategy which ignores the order of the relation embeddings:

$$sim_3(\mathbf{r_{ac_i}}; \mathbf{r_{c_i b}}, \mathbf{r_{xz_j}}; \mathbf{r_{z_j y}}) \qquad (6)$$
$$= \cos(\mathbf{r_{ac_i}} + \mathbf{r_{c_i b}}, \mathbf{r_{xz_j}} + \mathbf{r_{z_j y}})$$

# 7 Experiments

We empirically analyse our proposed strategies for solving hard analogy questions. We are specifically interested in the following research questions. (i) How suitable is the confidence degree (3) as a proxy for estimating the difficulty of an analogy question? (ii) How suitable are relation embedding chains for answering difficult analogy questions? (iii) What are the best ways for learning and using these relation embedding chains?

## 7.1 Experimental Setup

**Datasets** We evaluate our models on a number of different analogy question datasets. First, we use the 5 datasets that were used by Ushio et al. (2021b): the SAT dataset proposed by Turney et al. (2003); the U2 and U4 datasets which were obtained from an educational website; and reformulations of the Google (Mikolov et al., 2013) and BATS (Gladkova et al., 2016) datasets into the multiple-choice analogy question format. We obtained these datasets from the RelBERT repository[4]. We also use a reformulation of SCAN (Czinczoll et al., 2022b) into the multiple-choice analogy question format, which is available from the

same repository. Finally, we include the English version of E-KAR[5] (Chen et al., 2022). E-KAR contains questions questions involving word pairs and questions involving word triples. We only considered the former for our experiments. We use accuracy as the evaluation metric.

There are some important differences between these datasets. For instance, Google and BATS focus on morphological, encyclopedic, and in the case of BATS, lexical relations. SCAN focuses on scientific and creative analogies, requiring models to link relationships from one domain (e.g. the solar system) to relationships to a completely different domain (e.g. atoms). The other datasets focus on abstract conceptual relations, but they cover a range of difficulty levels (for humans): SAT is obtained from college entrance tests; U2 is aimed at children from primary and secondary school; U4 has difficult levels ranging from college entrance tests to graduate school admission tests; E-KAR was derived from Chinese civil service exams.

**Methods** The RelBERT repository on Huggingface contains a number of different RelBERT variants. We have used the model[6] trained using Noise Contrastive Estimation on SemEval 2012 Task 2 (Jurgens et al., 2012), as this variant outperforms the original variant from Ushio et al. (2021a).

For our implementation of *missing link prediction* we combined the top-250 neighbours (in terms of cosine similarity) according to the ConceptNet Numberbatch word embedding (Speer et al., 2017) with the top-250 neighbours according to GloVe (Pennington et al., 2014). For *semantic smoothing* we used to top-5 neighbours from GloVe.

In terms of baselines, our primary emphasis is on comparing the proposed methods with RelBERT, which is the state-of-the-art relation embedding model[7]. To provide additional context, we have also obtained results with GPT-4.[8] Following earlier work on using LLMs for solving analogy questions (Yuan et al., 2023), we ask the model to generate explanations (Wei et al., 2022) and include a few solved questions as part of the prompt. The

---

[4]https://huggingface.co/datasets/relbert/analogy_questions

[5]Available at https://ekar-leaderboard.github.io

[6]Available from https://huggingface.co/relbert/relbert-roberta-large-nce-semeval2012-0-400

[7]While Yuan et al. (2023) have reported higher results on some datasets using another fine-tuned RoBERTa model, their method does not learn relation embeddings but solves the task as a multiple-choice question answering problem. Moreover, their model was fine-tuned on the validation split of each benchmark, which makes the results incomparable.

[8]Details of our prompt can be found in the appendix.

| | SAT | | | | U2 | | | | U4 | | | | BATS | | | |
|---|---|---|---|---|---|---|---|---|---|---|---|---|---|---|---|---|
| Conf | RelB | Cond | Dir | GPT4 | RelB | Cond | Dir | GPT4 | RelB | Cond | Dir | GPT4 | RelB | Cond | Dir | GPT4 |
| [0.0, 0.25) | 40.7 | 51.9 | 37.0 | **59.3** | 17.6 | 35.3 | 52.9 | **76.5** | 26.9 | **46.2** | **46.2** | 38.5 | 32.1 | 46.7 | 55.5 | **90.5** |
| [0.25, 0.5) | 51.4 | 48.6 | **60.0** | 54.3 | 38.1 | 33.3 | **47.6** | 42.9 | 43.8 | 43.8 | 50.0 | **62.5** | 66.3 | 72.9 | 68.7 | **94.0** |
| [0.5, 0.75) | **78.0** | 71.0 | 67.0 | 65.0 | 65.9 | 70.5 | 61.4 | 70.5 | 59.3 | 52.3 | 48.8 | **61.6** | 68.0 | 69.5 | 67.0 | **87.3** |
| [0.75, 1.0] | **80.6** | 77.7 | 66.9 | 78.3 | **78.1** | 67.8 | 59.6 | 76.0 | 71.0 | 68.0 | 57.7 | **75.0** | 89.9 | 84.8 | 80.1 | **91.0** |

| | Google | | | | SCAN | | | | E-KAR | | | | Avg | | | |
|---|---|---|---|---|---|---|---|---|---|---|---|---|---|---|---|---|
| Conf | RelB | Cond | Dir | GPT4 | RelB | Cond | Dir | GPT4 | RelB | Cond | Dir | GPT4 | RelB | Cond | Dir | GPT4 |
| [0.0, 0.25) | 42.0 | 52.0 | 66.0 | **98.0** | 23.2 | 22.9 | 18.9 | **23.5** | 35.0 | **40.0** | 35.0 | 30.0 | 31.1 | 42.1 | 44.5 | **59.5** |
| [0.25, 0.5) | 50.0 | 50.0 | 57.1 | **100.0** | 26.7 | **28.3** | 27.4 | 20.4 | 42.3 | 34.6 | 19.2 | **50.0** | 45.5 | 44.5 | 47.2 | **60.6** |
| [0.5, 0.75) | 54.1 | 51.4 | 64.9 | **100.0** | 32.2 | **32.5** | 30.9 | 22.1 | 53.5 | **55.8** | **55.8** | 48.8 | 58.7 | 57.6 | 56.5 | **65.0** |
| [0.75, 1.0] | 93.0 | 88.1 | 84.9 | **98.9** | 33.6 | **35.5** | 35.0 | 31.4 | 52.3 | 50.8 | 41.5 | **60.0** | 71.2 | 67.5 | 60.8 | **72.9** |

Table 1: Results on analogy questions of different difficulty levels, across all datasets (accuracy). The best performing model per dataset and difficulty level is shown in bold.

| | SAT | U2 | U4 | BA | GO | SC | EK |
|---|---|---|---|---|---|---|---|
| GPT-4 | 70.3 | **71.9** | 68.8 | **90.8** | 99.0 | 23.5 | **51.3** |
| ChatGPT[†] | 65.8 | 68.9 | **70.8** | 90.6 | 97.8 | - | - |
| InstructGPT[†]$_{003}$ | 48.9 | 59.0 | 57.9 | 82.8 | 96.7 | - | - |
| RelBERT | 73.6 | 67.5 | 63.0 | 80.9 | 81.4 | 27.3 | 48.7 |
| Condensed | 70.6 | 62.7 | 60.9 | 79.2 | 78.6 | **27.9** | 48.1 |
| Direct | 63.8 | 58.3 | 54.4 | 75.7 | 79.2 | 25.7 | 40.9 |
| $Cond_{<0.25}$ | **74.5** | 68.9 | 64.1 | 82.0 | 82.4 | 27.2 | 49.4 |
| $Cond_{<0.5}$ | 74.2 | 68.4 | 64.1 | 82.7 | 82.4 | 27.6 | 48.1 |
| $Direct_{<0.25}$ | 73.3 | 70.2 | 64.1 | 82.7 | 83.8 | 25.6 | 48.7 |
| $Direct_{<0.5}$ | 74.2 | 71.1 | 64.8 | 82.9 | 84.4 | 25.7 | 44.8 |

Table 2: Overall results of the different models (accuracy). Results with † were taken from (Yuan et al., 2023).

results of GPT-4 should be interpreted with some caveats, however. For instance, it is well-known that the results of LLMs can be highly sensitive to the choice of the prompt, so it is likely that better results are possible. Moreover, it is possible that GPT-4 has seen some of the datasets during training, which could lead to inflated results. Besides GPT-4, we also include the results that were obtained by Yuan et al. (2023) using ChatGPT and InstructGPT$_{003}$ with chain-of-thought prompting.

## 7.2 Results

Table 1 shows our main results, focusing on four methods: RelBERT (*RelB*), condensed relation embedding chain comparison (*Cond*), direct relation chain comparison (*Dir*) and GPT-4. The results are broken down based on our confidence in RelBERT's prediction, as computed by (3). The overall results per dataset are summarised in Table 2. In this table, we also report the LLM results obtained by Yuan et al. (2023). We furthermore consider

four hybrid methods. The idea of these methods is to rely on RelBERT for the easy questions and on either *Cond* or *Direct* for the hard questions. For instance, $Cond_{<0.5}$ uses *Cond* for questions with difficulty levels below 0.5 (as estimated using (3)) and RelBERT for the others. A number of conclusions can be drawn.

**Confidence scores faithfully predict question difficulty** We can see that the performance of RelBERT is closely aligned with the predicted difficulty level. On average, across all datasets, the accuracy ranges from 31.1% for the hardest questions to 71.2% for the easiest questions. This pattern can be observed for all datasets. Moreover, the predicted difficulty level is also aligned with the performance of the other models. For instance, GPT-4 on average also performs best on the questions with high confidence values, especially for SAT, U4, E-KAR and SCAN. For Google and BATS, GPT-4 performs well throughout. This suggests that the confidence score (3) is successful in predicting intrinsic question difficulty, rather than merely predicting where RelBERT is likely to fail.

**Relation embedding chains are helpful for hard questions** Focusing on the performance for the hardest questions, with confidence levels in [0.0,0.25), we can see that *Condensed* outperforms RelBERT on all datasets, with the exception of SCAN (where the results are close). *Direct* outperforms in most datasets as well, but not in SCAN and SAT. For the questions with a difficult range in [0.25,0.50), *Direct* still outperforms RelBERT in most cases, with E-KAR now the only exception. For the questions with confidence level in the range [0.50,0.75) the performance of RelBERT is

| | SAT | U2 | U4 | BA | GO | SC | EK |
|---|---|---|---|---|---|---|---|
| Direct | 63.8 | 58.3 | 54.4 | 75.7 | 79.2 | 25.7 | 40.9 |
| $sim_2$ | **67.4** | 60.5 | 56.9 | **80.3** | **79.8** | **29.1** | 42.9 |
| $sim_3$ | 59.3 | 54.8 | 53.0 | 66.9 | 70.8 | 24.4 | 42.2 |
| CN + ss | 57.6 | 58.3 | 55.1 | 67.9 | 66.4 | 22.3 | 39.0 |
| CN + mlp | 58.8 | **64.5** | **59.3** | 77.4 | 78.0 | 22.8 | **44.8** |
| CN only | 56.7 | 57.5 | 57.4 | 71.2 | 62.0 | 21.4 | 41.6 |
| mlp only | 60.8 | 57.0 | 54.9 | 74.9 | 75.4 | 22.9 | 42.2 |
| CN types | 40.4 | 42.1 | 40.5 | 56.3 | 50.4 | 15.3 | 37.7 |

Table 3: Analysis of direct relation chain comparison.

| | SAT | U2 | U4 | BA | GO | SC | EK |
|---|---|---|---|---|---|---|---|
| Condensed | **70.6** | 62.7 | **60.9** | 79.2 | 78.6 | **27.9** | **48.1** |
| CN + ss | 69.4 | 63.2 | 60.4 | 76.9 | 74.6 | 27.3 | **48.1** |
| CN + mlp | 67.1 | 64.5 | **60.9** | **79.7** | **81.4** | 25.4 | **48.1** |
| CN only | 68.8 | 59.2 | 60.4 | 74.0 | 71.2 | 24.4 | **48.1** |
| mlp only | 65.9 | **67.1** | 59.3 | 78.0 | 80.2 | 25.1 | **48.1** |

Table 4: Analysis of condensed relation chain comparison.

generally quite similar to that of *Condensed* and *Direct*. Finally, for the easiest questions, RelBERT is clearly better, with the exception of SCAN. In accordance with these observations, in Table 2 we can see that the hybrid models generally outperform RelBERT, except for SCAN and E-KAR where their performance is similar. For SAT, three of the hybrid models outperform the state-of-the-art result of RelBERT. Finally, comparing *Condensed* and *Direct* there is no clear winner, with the former performing better for the easiest questions and the latter performing better for the hardest ones.

**GPT-4 performs best overall** It is also notable how similar the GPT-4 results are to the ChatGPT results obtained by Yuan et al. (2023). However, GPT-4 and ChatGPT do not provide relation embeddings and can thus not replace models such as RelBERT in many applications (e.g. for retrieval tasks). In Table 1, we can also see that *Condensed* and *Direct* can sometimes outperform GPT-4 on the hardest questions, namely for U4 and E-KAR. Interestingly, these are also the benchmarks with the highest intended difficulty level for humans. Moreover, for SCAN, which requires making suitable abstractions, GPT-4 generally underperforms.

### 7.3 Analysis

We now provide further analysis about the direct and condensed relation chain comparison methods. We also include a qualitative analysis to better understand the nature of relation embedding chains.

**Direct Relation Chain Comparison** Table 3 compares our default configuration (*Direct*), which uses $sim_1$ to measure the similarity between relation embedding chains, with a number of variants. First, the rows labelled $sim_2$ and $sim_3$ show the impact of replacing (4) by, respectively, (5) and (6) for measuring the similarity. The model based on $sim_2$ combines elements of the direct and condensed relation chain comparisons. Accordingly, we can see that its performance is typically in between that of these two methods. However, for BATS, Google and SCAN it actually outperforms both. The comparatively poor results for $sim_3$ show that the order of the relation embeddings matters.

Next, the table compares a number of variations in how the relation embedding chains themselves are constructed. As we discussed in Section 4, our main method combines ConceptNet with two augmentation strategies: semantic smoothing (*ss*) and missing link prediction (*mlp*). The rows labelled *CN + ss* and *CN + mlp* show the impact of only using one of these augmentation strategies. Note that our default configuration for *Direct* is *CN + mlp + ss*. Furthermore, the row labelled *CN only* shows the results when we only use ConceptNet paths, without applying either of the two augmentation strategies. Finally, the row labelled *mlp only* shows the results of a variant which only relies on the links predicted by the missing link prediction strategy, without using ConceptNet at all. As can be seen, the links from ConceptNet and those predicted using *mlp* lead to the best performance. In fact, the *CN + mlp* variant achieves the best results in several cases. Interestingly, *mlp* on its own already performs quite well, which shows that relation chains can be constructed even without the help of an external KG.

Finally, we also evaluate a variant which does not rely on relation embeddings (*CN types*). In this case, we only use ConceptNet paths for finding intermediate concepts. We then compute the compatibility $comp(a, b, x, y)$ as follows:

$$\sum_{i=1}^{u} \max_{j=1}^{v} \mathbb{1}[r_{ac_i} = r_{xz_j} \wedge r_{c_ib} = r_{z_jy}]$$

Here $r_{xy}$ is the ConceptNet relation type of the edge that connects $x$ and $y$, and $\mathbb{1}[\alpha]$ is 1 if the condition $\alpha$ is satisfied and 0 otherwise. As can be seen, this variant performs poorly. The contrast

between this method and *CN only* reflects the impact of (i) the noisy nature of ConceptNet and (ii) the fact that the fixed relation types are often less informative than relation embeddings.

**Condensed Relation Chain Comparison**  Table 4 compares our default approach based on condensed relation embedding chains (*Condensed*) with some variants. Note that our default configuration for *Condensed* is *CN + mlp + ss*. Specifically, as in Table 3, we analyse the impact of the semantic smoothing (*ss*) and missing link prediction (*mlp*) strategies. We can broadly observe the same patterns; e.g. we again find that *CN + mlp* achieves the best results in several cases and that it is possible to obtain competitive results without using a KG.

**Qualitative Analysis**  Table 7 shows some examples of informativeness scores for word pairs from ConceptNet. As these examples illustrate, concept pairs with high scores consistently have a clear relationship. Those with the lowest scores either do not have any obvious relationship, or they have a relationship which is not captured by their RelBERT embedding. As an example of this latter case, the table contains pairs that are linked by the "sounds like" (*chad : chat*) and "rhymes with" relation (*time : fine*).

Table 5 shows examples where RelBERT made an incorrect prediction while *Direct* picked the right answer. In each case, we also show the most influential intermediate concept for the query and answer pairs. Specifically, we find the intermediate concept $c$ for the query $(a, b)$ and the intermediate concept $z$ for the answer $(x, y)$ for which $sim_1(\mathbf{r_{ac}}; \mathbf{r_{cb}}, \mathbf{r_{xz}}; \mathbf{r_{zy}})$ is maximal. These examples illustrate some of the different roles that intermediate concepts can play. For instance, in the example from SAT, the intermediate worth *condemnable* makes it possible to characterise the pair *reprehensible:condemn* in terms of a near-synonym (*reprehensible:condemnable*) and a kind of morphological variation (*condemnable:condemn*). The example from U4 illustrates a case where the close similarity between two terms (*vernacular:regional*) is more easily recognised by RelBERT as a composition of two antonyms (*vernacular:national* and *national:regional*). The example from SCAN illustrates a case where the word pairs involved are only indirectly related.

The top half of Table 6 shows examples of questions that were predicted to be easy. As can be seen,

the word pairs involved have a clear and direct relationship. The bottom half of the table similarly shows examples of questions that were predicted to be hard. These examples are hard for different reasons. Some examples are challenging because they involve a clear but indirect relationship (e.g. *shopping:bank card*). Others involve rather abstract relations (e.g. *aloof:connected*). The example from Google reflects the fact that RelBERT was not trained on named entities.

## 8  Conclusions

Relations between concepts are typically modelled either as KG paths or as relation embeddings. In this paper, we have proposed a hybrid approach, which uses paths whose edges correspond the relation embeddings. This is useful to model relationships between concepts that are only indirectly related, and more generally for concept pairs whose relation embedding is unreliable. Our approach crucially relies on a classifier which can predict the informativeness of a RelBERT embedding. This classifier is used (i) to identify analogy questions where we should not rely on RelBERT embeddings; (ii) to select reliable RelBERT vectors for training a model for condensing relation embedding chains; and (iii) to identify informative paths beyond those in ConceptNet. We have relied on GPT-4 to generate training data for the informativeness classifier, which allowed us to address the lack of suitable existing datasets.

## Limitations

Our evaluation has focused on solving analogy questions, which are useful because they allow for a direct evaluation of different approaches for modelling relationships. While analogies play an important role in certain downstream tasks, we essentially regard this as in intrinsic evaluation task. It thus remains a matter for future work to analyse the usefulness of relation embedding chains in downstream tasks (e.g. retrieving relevant cases for case-based reasoning methods). From a practical point of view, using relation embedding chains is more involved than directly using RelBERT embeddings. To address this, the augmentation strategies can be applied offline, by creating a graph of related concepts. While the corresponding RelBERT embeddings can also be computed offline, storing millions of relation embeddings takes up a large amount of space. In certain applications, it may

| | Query | RelBERT | Correct | Interm. query | Interm. answer |
|---|---|---|---|---|---|
| SAT | reprehensible : condemn | depraved : admire | estimable : praise | condemnable | praiseworthy |
| U2 | blandishment : coax | surplus : squander | eulogy : praise | seduction | homage |
| U4 | vernacular : regional | budget : austere | fluctuation : irregular | national | normal |
| SCAN | processing : bug | memorize : mistake | thinking : mistake | computing | working |
| BATS | package : parcel | hieroglyph : cloth | flower : blossom | courier | grower |

Table 5: Examples of analogy questions which are incorrectly answered by RelBERT, while being correctly answered by *Direct*. For each question, we also show the intermediate concept $c$ for the query pair $(a, b)$ and the intermediate concept $z$ for the answer pair $(x, y)$ for which $sim_1(\mathbf{r_{ac}}; \mathbf{r_{cb}}, \mathbf{r_{xz}}; \mathbf{r_{zy}})$ is maximal.

| | Dataset | Query | Candidates |
|---|---|---|---|
| EASY | SAT | abbreviation:word | **abridge:book**, laminate: layer, inhibit:idea, expedite:mail, invoke:deity |
| | U2 | sewing:craft | **gasoline:fuel**, salt:food, fate:science, sunrise:art |
| | U4 | pragmatic:practical | **trivial:negligible**, irritating:pleasing, tenacious:faltering, opaque:translucent |
| | BATS | market:marketplace | **sofa:couch**, murder:clothes, lazy:help, shirt:button |
| | Google | hand:hands | **horse:horses**, swimming:swam, dollars:goats, dollar:mouse |
| | SCAN | earth:air | **sun:space**, planet:orbit, planet:sun, planet:elliptical, planet:space, planet:gravity, ... |
| | E-KAR | bird:wings | **fish:fin**, sheep:dog, locust:cicada, cattle:grass |
| HARD | SAT | shallow:depth | **apathetic:caring**, salty:ocean, cloudy:height, lurid:shock, pious:faith |
| | U2 | aloof:connected | **deliberate:accidental**, rigid:firm, ethereal:fleeting, logical:calculating |
| | U4 | fastidious:particular | **fanatical:enthusiastic**, edible:delicious, adamant:opposed, manipulative:masterful |
| | BATS | shirt:button | **tonne:kilogram**, pie:tripod, fridge:appliance, sonata:door |
| | Google | Yerevan:Armenia | **Zagreb:Croatia**, Azerbaijan:Denmark, Dushanbe:Tunis, boy:girl |
| | SCAN | money:effective | **time:efficient**, schedule:quick, time:schedule, time:quick, time:slow, ... |
| | E-KAR | shopping:bank card | **cooking:natural gas**, driving:steering wheel, running:sneakers, travel:navigator |

Table 6: Top: examples of analogy questions with a confidence in $[0.75, 1.0]$, i.e. questions predicted to be easy. Bottom: examples of analogy questions with a confidence in $[0, 0.25)$, i.e. questions predicted to be hard. In each case, the correct answer is shown in bold.

| Pair | Score |
|---|---|
| horse : pony | 0.99 |
| intermarriage : intramarriage | 0.99 |
| foresight : hindsight | 0.96 |
| addressable : unaddressable | 0.95 |
| balloon : popped | 0.52 |
| call : text | 0.50 |
| at one time : individually | 0.49 |
| fate : choice | 0.47 |
| farmer : slicker | 0.10 |
| moment : ages | 0.09 |
| time : fine | 0.01 |
| chad : chat | 0.01 |

Table 7: Examples of the predicted informativeness scores for concept pairs from ConceptNet.

thus be preferable to compute the required Rel-BERT embeddings only when they are needed.

**Acknowledgments** This work was supported by EPSRC grants EP/V025961/1 and EP/W003309/1.

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

## A  Generating Examples of Related Concept Pairs with GPT-4

Two concepts can be related in various ways, including, but not limited to, the following:

- Semantic relationship: This includes synonyms, antonyms, hyponyms, hypernyms, meronyms, holonyms, etc.

- Function or purpose, e.g., *key* and *lock*.

- Manner, way, or style: Concepts that describe the way in which another concept is accomplished, e.g., *limp* and *walk*.

- Symbol or representation: Concepts where one represents or symbolizes the other, e.g., *dove* and *peace*.

- Degrees of intensity: Concepts that describe different degrees of intensity of a particular situation, e.g., *shower* and *monsoon*.

- Cause and effect: Concepts where one causes the other or results from the other, e.g., *fire* and *smoke*.

- Sequence or hierarchy: Concepts that follow each other in a sequence or are organized hierarchically, e.g., *manager* and *employee*.

In order to identify a large list of such relation types, we asked GPT-4:

```
'''
Let A and B be two concepts in natural
    language. When can we say that A is related
    to B, and when is A not related to B?
'''
```

We identified approximately 100 relationships through this approach, and then again used GPT-4 to generate examples for each of these relationships, using prompts of the following form:

```
'''
A way two concepts can be related is

Tools and materials: Concepts where one is a
    tool or material used to create or interact
    with the other, e.g., 'paintbrush' and
    'paint.'

Now, generate 100 high-quality examples of such
    concept pairs.
'''
```

We prompted the model once for each relation type to maintain a balanced number of examples for each relationship. The generated examples were utilized to train our informativeness classifier.

## B  Additional Training Details

To train the composition model, we used 200,000 concept pairs from ConceptNet (version 5.6.0) with

| Dataset | #Quest. | #Cand. |
|---------|---------|--------|
| SAT | 337 | 5.0 |
| U2 | 228 | 4.3 |
| U4 | 432 | 4.2 |
| BATS | 1799 | 4.0 |
| Google | 500 | 4.0 |
| SCAN | 1616 | 74.5 |
| E-KAR | 154 | 4.0 |

Table 8: Overview of the considered datasets, showing for each dataset the number of questions in the test set (#Quest.) and the average number of options per question (#Cand.).

| Conf | SAT | U2 | U4 | BA | GO | SC | EK |
|------|-----|-----|-----|------|-----|-----|-----|
| $[0.0, 0.25)$ | 27 | 17 | 26 | 137 | 50 | 652 | 20 |
| $[0.25, 0.5)$ | 35 | 21 | 48 | 166 | 42 | 427 | 26 |
| $[0.5, 0.75)$ | 100 | 44 | 86 | 197 | 37 | 317 | 43 |
| $[0.75, 1.0]$ | 175 | 146 | 272 | 1299 | 371 | 220 | 65 |

Table 9: Number of analogy questions for each of the difficult ranges.

informativeness scores greater than 0.75. The concepts belonged to the English language only. We did not consider ConceptNet relations "/r/NotCapableOf", "/r/NotDesires", and "/r/NotHasProperty" for determining intermediate concepts. Our default augmentation strategy, i.e., CN+MLP+ss, identified approximately 50 intermediate concepts for each pair of concepts on average. We used Numberbatch *version 19.08* and Glove model *glove-wiki-gigaword-300* in our augmentation strategy. The concept pairs for which the strategy could not determine any path were not considered for training. 10% of the selected concept pairs were utilized for validation. We trained a function $\phi : \mathbb{R}^{1024} \times \mathbb{R}^{1024} \rightarrow \mathbb{R}^n$, where the validation set was used to choose the optimal dimension $n$ of the latent space. We observed in particular that high-dimensional latent spaces outperformed lower-dimensional ones, where we found $n = 81,920$ to be optimal. The Adam optimizer was used with a learning rate of 0.0025, and the model was trained for 10 epochs. It is important to note that we did not use validation splits from the analogy datasets to tune these hyperparameters, as we consider the unsupervised setting.

## C  Details about the Analogy Question Datasets

Some basic statistics about the considered datasets are shown in Table 8. Table 9 shows how many questions there are in each difficulty range, for each

of the datasets, where the difficulty is measured in terms of the confidence score (3) as before.

## D  Solving Analogy Questions using GPT-4

To solve analogy questions with GPT-4, we used the following prompt, which was inspired by (LearningExpress, 2002).

```
'''
Many standardized tests, including high school
    entrance exams, the SATs, civil service
    exams, the GREs, and others, use analogy
    questions to test both logic and reasoning
    skills and word knowledge. These questions
    ask test takers to identify relationships
    between pairs of words.

In order to solve analogy questions, you must
    first have a clear understanding of the
    words' definitions and then use that
    understanding to determine how the words
    are related. The key to solving an analogy
    question is to precisely describe the
    relationship between the pair of words and
    then apply the same relationship to
    determine which word completes the analogy.
    Most analogy questions rely on your ability
    to deduce the correct relationship between
    words and to draw logical conclusions about
    the possible answer choices.

The relationships that are found in analogy
    questions fall into several general types.

1) Part to Whole. In this type of question, a
    pair of words consists of a part and a
    whole. For example, spoke : wheel. A spoke
    is part of a wheel.

2) Type and Category. These questions use pairs
    of words in which one word is a specific
    type in a general category. For example,
    orange : citrus. An orange is a type of
    citrus.

3) Degree of Intensity. These questions test
    your ability to discern nuance of meaning
    among pairs of words. For example, shower :
    monsoon. A shower is light rainfall and a
    monsoon is heavy rainfall.

4) Function. These questions pair words that
    are related through function. For example,
    hammer : build. A hammer is used to build.

5) Manner. This type of analogy describes the
    manner, way, or style by which an action is
    accomplished. For example, shamble : walk.
    Shamble means to walk in an awkward manner.

6) Symbol or representation. These questions
    pair words in which one word is the symbol
    of the other. For example, dove : peace. A
    dove is a symbol of peace.

7) Action and significance. In this type of
    analogy one word describes an action and
```

| Conf | RelB | Cond | Dir | GPT4 |
|------|------|------|-----|------|
| $[0.0, 0.25)$ | 26.3 | 30.0 | 29.1 | 39.9 |
| $[0.25, 0.5)$ | 39.6 | 41.4 | 41.2 | 46.5 |
| $[0.5, 0.75)$ | 53.0 | 52.2 | 50.2 | 54.5 |
| $[0.75, 1.0]$ | 81.2 | 76.9 | 71.4 | 82.8 |

Table 10: Results for the main experiments, micro-averaged across all datasets.

| Conf | RelB | Cond | Dir | GPT4 |
|------|------|------|-----|------|
| $[0.0, 0.25)$ | 43.9 | 45.5 | 48.6 | 72.5 |
| $[0.25, 0.5)$ | 39.3 | 39.9 | 41.3 | 51.6 |
| $[0.5, 0.75)$ | 41.5 | 45.6 | 41.9 | 52.4 |
| $[0.75, 1.0]$ | 67.3 | 64.5 | 60.4 | 68.2 |

Table 11: Micro-averaged results, broken down according to the confidence predicted by the informativeness classifier that was trained on Conceptnet.

```
    the other word indicates the significance
    of the action. For example, cry : sorrow.
    To cry signifies sorrow

Analogy questions can also be used to test word
    knowledge and factual content. Word
    knowledge questions are generally pairs of
    synonyms or pairs of antonyms. Factual
    content questions demand a certain level of
    general knowledge, and cannot be deduced
    from the relationship alone.

Given the word pair, your aim is to choose the
    word pair from choices that is analogously
    most similar. Also, give an explanation.
    The explanation should be precise. I will
    show some examples then you will have to do
    it yourself.

Query = ['banana', 'peel']; Choices = [['egg',
    'crack'], ['carrot', 'uproot'], ['apple',
    'core'], ['bread', 'slice'], ['corn',
    'husk']]

Answer: choice number 4; Explanation: A banana
    has a peel that can be removed, and corn
    has a husk that can be removed.

Query = ['birds', 'wings']; Choices =
    [['moose', 'antlers'], ['camel', 'hump'],
    ['spider', 'legs'], ['alligator', 'tail'],
    ['cat', 'whiskers']]

Answer: choice number 2; Explanation: Birds
    have wings, and spiders have legs.

Query = ['berate', 'criticize']; Choices =
    [['goad', 'urge'], ['accuse', 'apologize'],
    ['regret', 'remember'], ['betray',
    'follow'], ['evaluate', 'praise']]

Answer: choice number 0; Explanation: To berate
    is to criticize, and to goad is to urge.

Now, answer the following questions:
'''
```

## E  Additional Results

Table 1 reports the macro-averaged accuracy, as a summary of the performance of the different methods across all datasets. In this macro average, each *dataset* carries the same weight. To complement this result, Table 10 instead shows the micro-averaged result, where each *analogy question* car-

ries the same weight, regardless of the dataset it appears in. The main conclusions remain the same as the ones we could draw based on the macro-average. One difference is that the spread in performance between the easiest and the hardest questions is even wider for the micro-averaged results.

For comparison, in Table 11 we also show the micro-averaged result, but in this case, we break down the results based on an informativeness classifier that was trained on examples from ConceptNet as positive examples. In contrast, for our main experiments, this informativeness classifier was trained on examples we obtained from GPT-4. As can be seen in Table 11, the informativeness classifier based on ConceptNet is far less successful in identifying easy and hard questions. For instance, GPT-4 actually achieves the best results on the questions that are predicted to be hardest.