# OpenReview forum: "Solving Hard Analogy Questions with Relation Embedding Chains"
_EMNLP/2023/Conference — EMNLP 2023 Main_

### Official Review · Reviewer_S1H7 · 2023-08-02

**Soundness:** 3

**Excitement:**

3: Ambivalent: It has merits (e.g., it reports state-of-the-art results, the idea is nice), but there are key weaknesses (e.g., it describes incremental work), and it can significantly benefit from another round of revision. However, I won't object to accepting it if my co-reviewers champion it.

**Missing References:**

None

**Paper Topic And Main Contributions:**

This paper proposes a method for solving analogy questions by considering paths between concepts and relation embeddings distilled from large language models. To do so, they first learn a binary classifier that predicts a confidence score for a relation embedding RelBERT to use it as a threshold for predicting links between concepts. To find a path between two concepts, they propose to use word similarities in addition to the links provided in a structured knowledge graph (e.g., ConceptNet). This is done by first finding the top-k nearest neighbour for a concept in a given pair using static word embeddings, then choosing those words with informative relation embeddings (high confidence score) with the concept. Experimental results on multiple datasets show that the proposed method performs well for hard analogy questions in which the confidence score for the answers using the RelBERT embedding is low.

**Questions For The Authors:**

- Which similarity measures (sim1, sim2 or sim3) is used for the results in Table1 for Direct method?
- The third baseline in equation 6 for measuring similarity between two relation chains is not motivated. Why did you decide to add the two relation embeddings? Why not concatenate? Subtracting?
- Table 3 did not show the result of the full proposed model (CN +ss+mlp).

**Reasons To Accept:**

- The novelty of using chains of relation embeddings to solve analogy questions.
- Extensive experiments that include analogy questions benchmarks and multiple baselines.
- The incorporation of relation embeddings and relational paths between a pair of concept to solve analogy questions is novel, as most related work focus on the two concepts in the given query ignoring the intermediate concepts that could lead to the targeted relation in a compositional way.
- Interesting analysis of the main results in Table 1 (with highlighted summaries).

**Reasons To Reject:**

- As the first step of missing link prediction is to find the semantically similar words y to a concept a, then the link is added if the relation embedding between an and y is informative. This step of adding relational links between concepts is questionable as it focuses on words that are semantically similar based on their attributes from static word embeddings. By doing so, we don't expect to get words that are related to a by some relations instead we got synonyms and semantically similar neighbours.
- The proposed method is weak in solving easy analogy questions. However, we expect a method that works well on hard cases to also perform well on easy cases. Does that mean for easy questions the relation between an and b are direct and the missing link prediction injects noisy paths?

**Reproducibility:**

4: Could mostly reproduce the results, but there may be some variation because of sample variance or minor variations in their interpretation of the protocol or method.

**Reviewer Confidence:**

5: Positive that my evaluation is correct. I read the paper very carefully and I am very familiar with related work.

**Typos Grammar Style And Presentation Improvements:**

- Rewrite the abstract to clarify some important concepts such as informative relation embeddings? hard analogy questions?
- Similarly, the last part of the introduction section is not well-motivated as you have to briefly explain your proposed approach/techniques. In addition, the analogy question (hard ones) are not explained. how do you classify a question to be hard or not? As this terminology is also used in the title, it is important to clarify it better in early sections.
- For better analysis, show the answers for the RelBERT and the proposed method for the shown easy/hard questions in Table 6.
- Clearly state the setting for the hybrid models shown in Table 2.
- It is better to explain the notations of Table 3 in its caption for quick access.
- The missing link prediction section from line 209 needs clarification. How do you add links from b to? Do you mean y must be connected to both a and b?
- In line 201, do you mean a-> x1 -> x2-> …->b not y?

---

> ### Author Rebuttal · Authors · 2023-08-25
>
> We thank the reviewer for the insightful comments. We now address the concerns raised by the reviewer one by one as below.
>
> Q1: As the first step of missing link prediction is to find the semantically similar words y to a concept a, then the link is added if the relation embedding between an and y is informative. This step of adding relational links between concepts is questionable as it focuses on words that are semantically similar based on their attributes from static word embeddings. By doing so, we don't expect to get words that are related to a by some relations instead we got synonyms and semantically similar neighbours.
>
> A1: When using static word embeddings, some of the neighbours we find are indeed synonyms or morphological variations, but we also often find words that are related in different ways. Note that we check for informative relation embeddings among the top 500 neighbors, a number we find sufficient to go beyond just synonyms and morphological variations. For instance, some links that are not present in ConceptNet and have been added using this approach include the following:
>
> (“dog”, “owners”): Reflects the relationship that dogs can have owners.
> ("cashier", "grocery store"): Cashiers have a key role in grocery stores.
> ("helium", "noble gases"): Helium belongs to the category of noble gases.
> ("drug trafficking", "illegal"): Drug trafficking is illegal.
> (“disinfectant”, “sterilization”): Disinfectants are used for sterilization.
>
> Such links clearly enrich ConceptNet in a non-trivial way. To make this clearer in the paper, we will add a qualitative analysis about the kinds of semantically related words that are found by our strategy.
>
>
> Q2: The proposed method is weak in solving easy analogy questions. However, we expect a method that works well on hard cases to also perform well on easy cases. Does that mean for easy questions the relation between an and b are direct and the missing link prediction injects noisy paths?
>
> A2: Easy questions indeed typically require modelling direct relationships, which can be modelled well using RelBERT relation embeddings (see Table 6 for examples of easy and hard questions). In fact, the proposed methods are quite robust in the presence of noisy paths. The main limitation is rather that no suitable intermediate concepts may be found in some cases. In other words, the effectiveness of the method mostly depends on the quality of the most informative path.
>
> It is indeed quite intuitive that we should only try to model relationships using intermediate concepts (i.e. using relation embedding chains) in cases where the relationship cannot be modelled directly. One of our most significant findings is that we can identify those cases using the proposed informativeness classifier. As our results in Table 2 show, this allows hybrid methods such as Cond_{<0.25} to perform well on both easy and hard questions.
>
>
> Q3: Which similarity measures (sim1, sim2 or sim3) is used for the results in Table1 for Direct method?
>
> A3: We used “sim1” for the Direct method. We will clarify this in the paper.
>
>
> Q4: The third baseline in equation 6 for measuring similarity between two relation chains is not motivated. Why did you decide to add the two relation embeddings? Why not concatenate? Subtracting?
>
> A4: We introduced the third baseline, referred to as "sim3," to specifically analyse the importance of modeling the order of relations in a chain. Unlike "sim1" and "sim2," "sim3" does not take into account the order of relations. We find that, as a result, it performs poorly in comparison.
>
>
> Q5: Table 3 did not show the result of the full proposed model (CN +ss+mlp).
>
> A5: Our default configuration, which is labeled 'Direct' in the table, is CN + ss + mlp. We will clarify this in the paper.
>
>
> Q6: Regarding typos and presentation improvements.
>
> A6: We are grateful for the suggestions and will address them in the paper.
>
> Indeed, "y" is a typo; we meant 'a -> x1 -> x2 -> ... -> b.'
>
> As suggested, we will improve the 'missing link prediction' section for greater clarity. Regarding the question of whether 'y' must be connected to both 'a' and 'b': the answer is no, it may not be. Specifically, a path 'a -> y -> b' exists if 'a -> y' is the predicted link and 'y -> b' is not a predicted link but exists in ConceptNet."

---

### Official Review · Reviewer_yTYV · 2023-08-05

**Soundness:** 4

**Excitement:**

4: Strong: This paper deepens the understanding of some phenomenon or lowers the barriers to an existing research direction.

**Paper Topic And Main Contributions:**

The paper first discusses the pros and cons of using vectors, instead of a fixed set of relation types, to represent the relations between entities.

In this paper, a new approach is proposed to combine the good perspectives of using vectors and fixed relation types. The authors propose to use relation embedding chains.

The proposed method is built on the RelBERT, where the relation embedding is first computed from any pair of words. Then they train a classifier for related/unrelated concepts based on the samples generated from the large language models, namely GPT4. A trained classifier can be used to give the informativeness of the embedding.


The generated relation embedding tasks can be used for answering hard analogy questions. The proposed method achieved state-of-the-art performance on the analogy questions.


**Questions For The Authors:**

I do not have specific questions for the authors.

**Reasons To Accept:**

The idea of using word chains to represent open relations is very novel and interesting.

The experiment is extensive and persuasive.


**Reasons To Reject:**

The impact of learning a good relation embedding is rather limited in the current research environment.  However, this is still a very good idea worth the community to know, especially for the people who are working on constructing and leveraging knowledge graphs.


**Reproducibility:**

5: Could easily reproduce the results.

**Reviewer Confidence:**

4: Quite sure. I tried to check the important points carefully. It's unlikely, though conceivable, that I missed something that should affect my ratings.

---

> ### Author Rebuttal · Authors · 2023-08-25
>
> We thank the reviewer for the motivating comments. We now address the concern raised by the reviewer.
>
> Q1: The impact of learning a good relation embedding is rather limited in the current research environment.
>
> A1: We interpret "current research environment" to refer to large language models (LLMs) such as GPT-4. While it is clear that GPT-4 performs very well on analogy questions, our experiments also show that in some cases we can still get better results using relation embedding chains. More fundamentally, learning representations using smaller/efficient models remains important in various applications. For instance, relation embeddings have an obvious role to play in applications that currently rely on (commonsense) knowledge graphs, such as recommendation (where existing models cannot simply be replaced by an LLM). Representations also matter for information retrieval, and we believe that relation embeddings have an important role to play in this area as well.

---

### Official Review · Reviewer_zL7T · 2023-08-05

**Soundness:** 4

**Excitement:**

3: Ambivalent: It has merits (e.g., it reports state-of-the-art results, the idea is nice), but there are key weaknesses (e.g., it describes incremental work), and it can significantly benefit from another round of revision. However, I won't object to accepting it if my co-reviewers champion it.

**Paper Topic And Main Contributions:**

This paper proposes a concept relation representation method, which views relations as paths, and labels them with relation embedding, which become relation embedding chains.

**Reasons To Accept:**

It introduced a training model to learn relation embeddings by predicting the similarity of RelBERT embedding with relation embedding chains. Experimental results on hard analogy question shows the benefit of this representation method.

**Reasons To Reject:**

Besides GPT-4, the proposed method is only compared with RelBERT on hard analogy questions. How about other models, and other tasks, such as relation classification.

**Reproducibility:**

3: Could reproduce the results with some difficulty. The settings of parameters are underspecified or subjectively determined; the training/evaluation data are not widely available.

**Reviewer Confidence:**

2: Willing to defend my evaluation, but it is fairly likely that I missed some details, didn't understand some central points, or can't be sure about the novelty of the work.

---

> ### Author Rebuttal · Authors · 2023-08-25
>
> We thank the reviewer for the insightful comments. We now address the concern raised by the reviewer.
>
> Q1: Besides GPT-4, the proposed method is only compared with RelBERT on hard analogy questions. How about other models, and other tasks, such as relation classification.
>
> A1:  We chose to compare our method with RelBERT because it is the state-of-the-art model for the datasets we considered. Moreover, our specific focus is on showing the benefit of relation embedding chains, and since we use RelBERT for modelling such chains, comparing with RelBERT allows us to directly evaluate this aspect.
>
> We did not consider relation classification because existing benchmarks focus on rather coarse-grained relations (where the benefit of using RelBERT embeddings is more limited), and because, by construction, they only involve word pairs that have a direct relationship.

---

### Meta-Review · Area_Chair_fvXx · 2023-09-13

**Recommendation:** 4

**Metareview:**

This paper presents an application of relation embeddings as a method for finding relation chains between concepts in the ConceptNet knowledge graph, implemented also for finding analogies, the task on which the method is evaluated.

The reviewers agree that the paper presents a novel method in a clear way, with some expressing excitement at its elegant nature and impressive results over the benchmarks. All concerns were addressed by the author responses, which were acknowledged by the reviewers. If accepted, I recommend the authors evaluate the benchmarks on at least one more competing method as proposed by reviewer zL7T, and add a short discussion featuring examples of nearest-neighbor words to alleviate the first concern raised by reviewer S1H7.

As a postscript, although I am not an official reviewer and have not carefully read the paper, but on a quick skim I noticed that there is no reference to literature on multi-hop relations and models, e.g. https://aclanthology.org/2020.acl-main.412/ (and much preceding work in ACL and non-ACL venues). This seems fairly relevant, and I hope the authors can look into this literature as well.

---

### Decision · Program_Chairs · 2023-10-07

**Decision:**

Accept-Main

**Comment:**

This paper presents an application of relation embeddings as a method for finding relation chains between concepts in the ConceptNet knowledge graph, implemented also for finding analogies, the task on which the method is evaluated.

The reviewers agree that the paper presents a novel method in a clear way, with some expressing excitement at its elegant nature and impressive results over the benchmarks. All concerns were addressed by the author responses, which were acknowledged by the reviewers. If accepted, I recommend the authors evaluate the benchmarks on at least one more competing method as proposed by reviewer zL7T, and add a short discussion featuring examples of nearest-neighbor words to alleviate the first concern raised by reviewer S1H7.

As a postscript, although I am not an official reviewer and have not carefully read the paper, but on a quick skim I noticed that there is no reference to literature on multi-hop relations and models, e.g. https://aclanthology.org/2020.acl-main.412/ (and much preceding work in ACL and non-ACL venues). This seems fairly relevant, and I hope the authors can look into this literature as well.